# Saudi Consensus Recommendations on the Management of Multiple Sclerosis: Symptom Management and Vaccination

Ibtisam A. Al Thubaiti [1,†], Mona M. AlKhawajah [2,†], Norah Al Fugham [3], Dema A. Alissa [4], Ahmed H. Al-Jedai [4], Yaser M. Al Malik [5], Mousa A. Almejally [6], Hajer Y. Al-Mudaiheem [4], Bedor A. Al-Omari [7], Hessa S. AlOtaibi [8], Rumaiza H. Al Yafeai [9], Mohammed A. Babakkor [10], Reem F. Bunyan [11], Edward J. Cupler [12], Mohammed Hakami [13], Hanaa M. Kedah [6], Seraj Makkawi [14,15,16], Leena H. Saeed [17], Jameelah A. Saeedi [18], Eslam Shosha [19] and Mohammed A. Al Jumah [20,*]

1   Department of Neurology, King Fahad Military Medical Complex Dhahran, Dhahran 31932, Saudi Arabia
2   Department of Neurosciences, King Faisal Specialist Hospital and Research Center, Riyadh 11564, Saudi Arabia
3   Neurology Department, King Faisal Specialist Hospital and Research Center, Riyadh 11564, Saudi Arabia
4   Deputyship of Therapeutic Affairs, Ministry of Health, Riyadh 12382, Saudi Arabia
5   College of Medicine, King Saud Bin Abdulaziz University for Health Sciences, Riyadh 14611, Saudi Arabia
6   Department of Neurology, Hera General Hospital, Makkah 9364, Saudi Arabia
7   Department of Pharmacy, Prince Sultan Military Medical City, Saudi Society of Clinical Pharmacy (SSCP) Makkah Al Mukarramah Rd, As Sulimaniyah, Riyadh 12233 Saudi Arabia
8   Neurology Unit, Department of Medicine, King Fahad Hospital, Jeddah 23325, Saudi Arabia
9   Department of Neurology, Psychiatry & Psychology My Clinic International Medical Co., Jeddah 23617, Saudi Arabia
10  Department of Neurology, King Abdullah Medical City, Makkah Al-Mukarramah 24246, Saudi Arabia
11  Department of Neurology, King Fahad Specialist Hospital Dammam, Dammam 32253, Saudi Arabia
12  Department of Neurosciences, King Faisal Specialist Hospital and Research Center, Jeddah 23431, Saudi Arabia
13  Neurology Unit, King Fahad Central Hospital, Jazan 82666, Saudi Arabia
14  College of Medicine, King Saud bin Abdulaziz University for Health Sciences, Jeddah 22384, Saudi Arabia
15  King Abdullah International Medical Research Center, Jeddah 23816, Saudi Arabia
16  Department of Medicine, Ministry of the National Guard-Health Affairs, Jeddah 23235, Saudi Arabia
17  Department of Pharmacy, King Fahad Medical City, Riyadh 12231, Saudi Arabia
18  Department of Neurology, King Abdullah Bin Abdulaziz University Hospital, Riyadh 11564, Saudi Arabia
19  Division of Neurology, Department of Medicine, McMaster University, Hamilton, ON L8S 4L8, Canada
20  Department of Neurology, King Fahad Medical City, Riyadh 12231, Saudi Arabia
*   Correspondence: jumahm@gmail.com
†   These authors contributed equally to this work.

**Abstract:** This article deals with recommendations on the management of symptoms of MS and on the provision of vaccinations in patients receiving disease-modifying therapies (DMTs). Symptoms of MS, such as fatigue, depression, urinary symptoms, spasticity, impairment of gait, and sexual dysfunction, are common in this population. Recognizing and addressing these symptoms is key to maintaining the quality of life of people with MS. Vaccination status should be reviewed and updated prior to initiation of DMTs. In general, vaccination should be avoided for variable periods after the initiation of some DMTs. Live attenuated vaccines are contraindicated and should be considered on a case-by-case basis. These consensus recommendations will present the best practices for vaccination in Saudi Arabia before, during, and after the COVID-19 pandemic. The recommendations will be updated periodically and as needed as new evidence becomes available.

**Keywords:** Saudi consensus; multiple sclerosis; symptom management; vaccinations; patients

## 1. Introduction

Through this review article, we aim to provide clinicians with the latest recommendations for managing symptoms of MS and for providing vaccinations to patients receiving disease-modifying therapies (DMTs). To develop the below recommendations, a group of neurologists, nurses specialized in MS, neuroradiologists, and pharmacists gathered to discuss, review and agree upon the latest relevant guidelines. This paper is meant to be complementary to the previously published manuscript from the same authors on DMTs entitled: "Saudi Consensus Recommendations on the Management of Multiple Sclerosis: Disease-Modifying Therapies and Management of Relapses" [1].

### 1.1. Symptoms of Multiple Sclerosis

Multiple sclerosis (MS) causes a range of symptoms that affect multiple aspects of physical and psychological functioning, with impacts ranging from distressing to disabling [2]. Accordingly, the appropriate diagnosis and management of these symptoms is a key element of the routine care of people with MS and has been described as an unmet clinical need in this area [3]. This article provides an overview of the burden and management of common and frequently distressing symptoms of MS, including lower urinary tract symptoms (LUTS), bladder dysfunction, paroxysmal symptoms, spasticity, impairment of gait, and sexual dysfunction. Other symptoms of MS that are often missed include depression, fatigue, and cognitive impairment. These common symptoms could be present at any time during the disease course, and they may fluctuate in intensity. Some symptoms may worsen during relapse. Accordingly, the management recommendations relate to MS in general, but the need for intervention may be particularly acute during a relapse, especially if its duration is prolonged.

### 1.2. Multiple Sclerosis and Vaccination

Treatment with some DMTs increases the risk of infections, including latent or opportunistic infections, such as varicella zoster or tuberculosis (TB) [4–6]. Encouraging patients to receive vaccination for these (and other) conditions is therefore important, especially in regions where diseases such as TB remain endemic [7,8]. All disease-modifying treatments (DMTs) for MS modulate the immune system in some way, with the potential to reduce the effectiveness of vaccination or to increase the risk of a live vaccine precipitating the infectious disease it was meant to prevent [9]. This article also presents recommendations on the appropriate timing and delivery of vaccinations for people under treatment with DMTs for MS.

## 2. Management of Common and Troublesome Symptoms of Multiple Sclerosis

### 2.1. Fatigue

Fatigue in MS has not been defined precisely but has been described by different authors as an overwhelming sense of exhaustion and lack of energy that is often not related to sadness or weakness and difficulty starting or maintaining voluntary effort (see the review by Mills and Young [10] for the sources of these definitions). Other studies have investigated the role of the thalamic network in the pathophysiology of MS-related fatigue [11]. Fatigue is one of the most common symptoms of MS and is reported by almost 80% of patients at some time [2,12]. Fatigue tends to be more common among patients with a longer duration of MS [2]. The majority of patients report fatigue as the worst symptom of MS they experience [12].

The causes of fatigue in MS are multifactorial and overlapping, and a multifaceted approach to evaluation and management is required. A clinical decision-making flowchart for managing fatigue in people with MS is shown in Figure 1 [12]. Before resorting to pharmacological therapy, it is important to evaluate the patient carefully and to identify and manage any factors unrelated to MS that contribute to fatigue, such as side effects of concomitant medications, anxiety disorders (particularly depression, which is commonly encountered among people with MS (see Figure 1 below), or sleep disturbances. Non-

pharmacologic approaches that may be of benefit include improving the diet, regular exercise, and management of daily tasks and the patient's ergonomic environment to conserve energy.

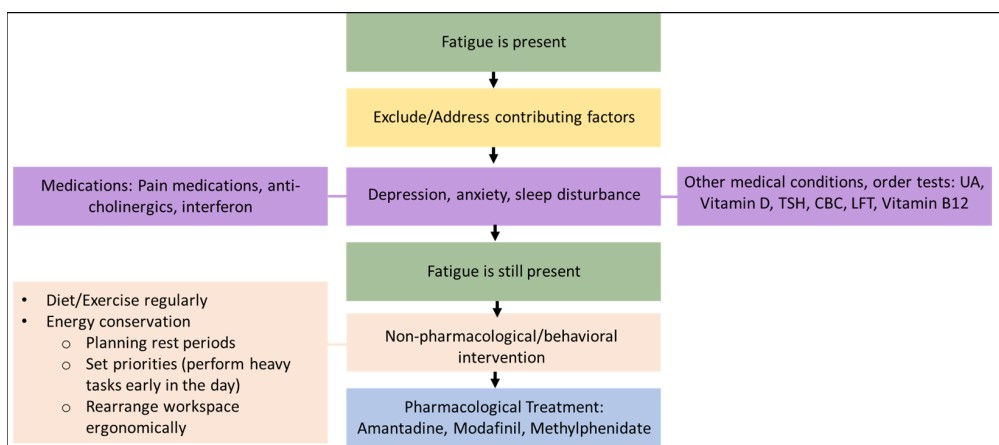

**Figure 1.** Management of fatigue in people with MS.

The figure was adapted with permission from the article by Khan F, Amatya B, Galea M. Management of fatigue in persons with multiple sclerosis. Front. Neurol. **2014**, *5*, 177. Copyright © 2023 Khan, Amatya and Galea.

The following medications may be considered for the management of fatigue in MS patients.

Amantadine has been shown to improve fatigue in seven randomized controlled trials (RCTs) and is recommended as a first-line treatment (100 mg twice daily) [13]. Although this treatment can be considered to be evidence-based, it should be noted that these RCTs were relatively small and of short duration.

Modafinil: A meta-analysis of five randomized trials demonstrated a significant improvement in one measure of MS fatigue (the Modified Fatigue Impact Scale) but with no effect on the fatigue severity scale [14]. Modafinil may be considered a second-line treatment for fatigue in patients with MS. Moreover, extended-release amantadine was shown no changes in fatigue levels in a double-blind, placebo-controlled, 4-week study [15].

Methylphenidate has been used for fatigue management in a variety of conditions. However, there are currently no trials supporting its use in people with MS [16].

### 2.2. Depression

The prevalence of depression is higher in people with MS than in the general population, with a lifetime prevalence as high as approximately 50% and an annual incidence approximately three-fold higher than that in the general population [17]. A large registry in the United States of America (USA) found a self-reported prevalence of depression of any severity in about three-quarters of people with a duration of MS of 10 years or more, with depression of moderate or higher severity reported by about one-third of this population [2].

Internationally validated, evidence-based questionnaires, such as the Patient Health Questionnaire [18] and the Beck Depression Inventory [19], are available for diagnosing depression. Psychoeducational approaches and subsequent follow-ups may be effective for mild-to-moderate depression of recent onset. For more severe or prolonged depression, evidence-based antidepressants may be prescribed only by experienced neurologists. It is also recommended to refer patients to psychiatry in the case of psychotic symptoms associated with depression. A referral for psychiatric support should also be considered for patients with clear signs of depression.

### 2.3. Cognitive Impairment

Cognitive impairment affects 40–65% of people with MS, with the exact prevalence depending on the classification [20]. Cognitive domains most frequently affected are episodic memory, attention, information processing speed, and executive function. When assessing cognitive impairment, compounding factors that need to be eliminated include depression, anxiety, fatigue, sleep disturbance (MS symptom, obstructive sleep apnea, Restless legs syndrome), and medications (antispastics, opioids, and some agents for neuropathic pain) [21]. The United States National MS Society has recommended that the Symbol Digit Modalities Test (or an equivalent validated test for cognition) should be administered at least annually to patients with MS (aged ≥8 years) in order to confirm and quantify any cognitive deficit, to follow the course of cognitive function over time, to detect new issues with cognition, and to evaluate the effects of treatments [22].

Strategies to improve cognition that may be recommended include conservative measures (diaries and calendars), regular physical exercise, and regular social contact [23]. While no pharmacologic therapy is currently recommended for the management of cognitive impairment in patients with MS, there are serious efforts to create evidence-based guidelines for cognitive rehabilitation for MS patients [24].

### 2.4. Lower Urinary Tract Symptoms/Bladder dysfunction

The prevalence of LUTS is estimated as high as 80–90% in patients with MS. Detrusor overactivity includes urgency, nocturia, frequent urination, and incontinence as the most common presentation [4].

MS is a leading cause of LUTS/bladder dysfunction among patients with neurological disorders, and this has a significant negative impact on quality of life [25]. Urinary symptoms are broadly categorized into failure to storage due to overactive bladder (urgency, frequency, nocturia, or urge incontinence), urinary retention due to failure of voiding (hesitancy, double voiding, straining, or sensation of incomplete emptying), or a combination of both [25].

Not only do LUTS reduce the quality of life of patients with MS [25], but they lead to urological complications like pyelonephritis and kidney stones, among others [4]. In fact, urological complications are one of the most common causes of hospitalization among MS patients [26].

First-line management may be initiated in neurological practice, but early referral to a urology practice should be considered if any of the following red flags are present: stress urinary incontinence, hydronephrosis, renal impairment, recurrent urinary tract infection (UTI), hematuria, suspected concomitant urologic pathology (for example., prostate enlargement), pelvic pain, or symptoms refractory to first-line treatment [25].

Conservative management strategies may be considered, including fluid restriction at night, scheduled voiding, avoidance of bladder irritants (for example., caffeine, tobacco, alcohol, carbonated beverages, chilli peppers, citrus fruits, and vitamin C supplements), and pelvic floor muscle training (PFMT) [25]. If this is ineffective, patients with overactive bladder and a postvoid residual volume <100 mL may be treated with anticholinergics. The following anticholinergics may be considered as first-line options: anticholinergics (oxybutynin, tolterodine, solifenacin, and trospium) or alpha antagonists (tamsulosin) [25] (see Table 1). Note that caution is recommended with the use of anticholinergic medications, especially in elderly patients, as they may cause central nervous system adverse effects to be monitored by the treating physician should they arise; these include headache, impaired cognition/memory, anxiety, insomnia, and behavioral disturbances. For this reason, dementia is a contraindication to the use of these medications. Other side effects caused by anticholinergics to be monitored include those affecting the peripheral nervous system, such as hyperthermia, constipation, blurred vision, and narrow-angle glaucoma. Therefore, it is also contraindicated to use these medications in patients with a history of glaucoma [27]. Second-line treatments should be initiated in urological practice and may include intravesical botulinum toxin injection [25]. In cases of voiding issues, alpha-blockers

can be used in addition to other strategies such as clean intermittent self-catheterization, indwelling catheters, percutaneous and transcutaneous tibial nerve stimulation, or surgical intervention [25]

**Table 1.** Medications used for the management of lower urinary tract symptoms/bladder dysfunction.

| Medication | Dose | Mechanism of Action | Side Effects |
|---|---|---|---|
| Oxybutynin | 5 mg twice to three times daily | Anti-cholinergic (anti-muscarinic M2-M3) causing smooth muscle relaxation | Dry mouth, constipation, difficulty in urination or retention, drowsiness, and rarely delirium |
| Tolterodine | 1–2 mg twice daily | New generation anti-cholinergic (anti-muscarinic) causing smooth muscle relaxation | Headache, dry mouth, constipation, difficulty in urination or retention and dizziness |
| Solifenacin | 5–10 mg daily | New generation anti-cholinergic (selective anti-muscarinic M3) causing smooth muscle relaxation | Dry mouth, less constipation, difficulty in urination or retention |
| Trospium | 20 mg twice daily | New generation anti-cholinergic (selective anti-muscarinic M3) causing smooth muscle relaxation | Dry mouth, less constipation, difficulty in urination or retention. |
| Tamsulosin | 0.4 mg daily | Blocks alpha 1 receptors on the urethral sphincter, decreasing the resistance on bladder smooth muscles. | Headache, orthostatic hypotension, and retrograde ejaculation in men. |

LUTS are likely the most common cause of hospitalization [28].

*2.5. Bowel Dysfunction*

Bowel dysfunction is a common problem affecting between 39–73% of MS patients [29]. Constipation is seen in nearly half of the patients, diarrhoea to a lesser extent (32%) and rarely fecal incontinence (2.5%) [29,30]. The treatment of bowel dysfunction remains unsatisfactory.

The causes of constipation are multi-factorial due to spinal cord lesions, decreased mobility or concomitant drug use (anti-cholinergic, narcotics, anti-convulsant, and muscle relaxant).

Treating constipation starts with physical exercise and dietary modification (high fiber diet and increased water intake) and eliminating causative drugs when possible.

Luxates can be used when necessary and include; psyllium, lactulose, polyethylene glycol, and bisacodyl. Other second-line options include prucalopride, transanal irrigation (TAI) or sacral neuromodulation, which require referral to gastroenterology or urology. The mode of action and side effects are mentioned in Table 2.

**Table 2.** Modalities used for management of bowel dysfunction symptoms.

| Drug/Modality | Mechanism of Action | Side Effects |
|---|---|---|
| Psyllium (dietary fiber) [29] | Bulking agent | Abdominal bloating |
| Lactulose [29] | Osmotic agents | Abdominal bloating and cramps |

**Table 2.** *Cont.*

| Drug/Modality | Mechanism of Action | Side Effects |
|---|---|---|
| Polyethylene glycol [29] | Osmotic agents | Nausea and abdominal bloating |
| Bisacodyl [29] | Stimulate enteric neurons | Nausea, diarrhoea, and cramps |
| Bisacodyl suppositories [29] | Rectal stimulants | |
| Prucalopride [29] | Selective 5-HT4 agonist; increases GI motility | Headache, nausea, abdominal pain, and diarrhoea. |
| Loperamide [29] | Anti-diarrheal; opioid-receptor agonist | Constipation, dizziness, nausea, cramps. |
| Transanal irrigation (retrograde irrigation) [29] | A device consists of a rubber catheter that is inserted in the rectum, with a balloon that is inflated to keep it in place and create a seal. Water is irrigated, and when the catheter is removed, a bowel action is obtained. Can be used for both constipation and fecal incontinence. | Requires a trained healthcare professional. Leaking of the remaining irrigated water. Rarely, perforation. Avoided in patients with previous pelvic surgeries. |
| Sacral neuromodulation [30] | Stimulation of the S2–S3 nerve roots or tibial nerve. Used for fecal incontinence | Limits MRI use. Efficacy on bowel dysfunction is not well established. |

Treatment of diarrhoea and incontinence starts with conservative therapy with antidiarrheals (loperamide), a special diet, and biofeedback if failed referral to gastroenterology or urology for TAI or sacral neuromodulation is advised (see Table 2).

### 2.6. Sexual Dysfunction

Sexual dysfunction is present in 50–80% of patients with MS, especially in men with a higher Expanded Disability Status Scale (EDSS). In women, sexual dysfunction is often secondary to fatigue [31]. Possible contributing factors should first be addressed, for example, medications for anxiety disorders (tricyclic antidepressants, selective serotonin reuptake inhibitors) and MS-related fatigue, depression, anxiety, and spasticity. A referral to urology or gynecology is also warranted [32]. Sildenafil has been shown to be effective for sexual dysfunction for men with MS; however, its benefits were limited to improvement in lubrication among women with MS. [33,34]. MS or its treatments do not seem to affect male fertility [35]. Similarly, MS in females of childbearing age does not appear to affect their fertility [36]. However, data on the topic of fertility in both sexes is still sparse and non-conclusive.

### 2.7. Paroxysmal Symptoms

Paroxysmal symptoms can include a wide variety of transient and stereotyped symptoms that can be sensory or motor lasting between 1–90 s [37]. The most common paroxysmal symptoms are trigeminal neuralgia and Lhermitte's sign. Tonic spasms (more with Transverse myelitis (TM) related to Neuromyelitis Optica (NMO) are less common. Uhthoff's symptoms (worsening of MS during an increase in temperature) may also present as paroxysmal symptoms [25].

No randomized trials have been conducted to guide therapy. However, for the management of tonic spasms, carbamazepine may be considered as the first-line therapy (low dose of approximately 200 mg two times a day) [38]. Other options for the management of tonic spasms include oxcarbazepine, gabapentin, and lacosamide [37]. Carbamazepine may

also be considered for the management of trigeminal neuralgia. Lhermitte's symptoms may be treated with amitriptyline, pregabalin, gabapentin, or duloxetine [37].

### 2.8. Spasticity

The initial treatment involves stretching exercises and rehabilitation. Stretches should be held for at least 30 to 60 s, and patients should be counseled to stretch twice daily. First-line pharmacologic options to manage spasticity include baclofen, tizanidine, or gabapentin, in addition to physiatry consultation or physical therapy. Second-line pharmacologic options include diazepam or dantrolene. For spasticity that is not generalized or that is considered local, botulism toxin injections and intrathecal baclofen pumps are treatment options [39]. Nabiximols are also safe to use to manage spasticity in patients that failed other treatments [40,41].

### 2.9. Gait Impairment

Gait impairment in MS is multifactorial and includes spasticity, weakness, fatigue, and sensory dysfunction. The pillar of gait impairment management is physical and occupational therapy. Dalfampridine is the main pharmacologic therapy to improve walking in MS (and the only Food and Drug Administration-approved treatment) in approximately one-third of patients [42]. The 25-foot walk test should be performed before and several weeks after starting the medication. Dalfampridine should be discontinued after a 4-week trial if there is no improvement in the 25-foot walk test.

### 2.10. Dysphagia

It is estimated that around 30–40% of MS patients suffer from Dysphagia or difficulty swallowing [43]. It could appear in adults with mild disability levels (EDSS 2-3) and is more prevalent in patients who are moderately or severely impaired (EDSS 8-9) [44]. Given the risk of aspiration pneumonia [45], an appropriate assessment of dysphagia through history-taking is necessary [46]. Video fluoroscopy or fiberoptic endoscopic assessment of swallowing is often used for definitive diagnosis [47]. Management of dysphagia requires the cooperation of a multidisciplinary team, including neurologists, nurses, speech and occupational therapists, otolaryngologists, and dieticians, to ensure adequate and safe nutrition.

## 3. Vaccination in People with Multiple Sclerosis

### 3.1. General Considerations Regarding Vaccination

Patients may be concerned that vaccinations could trigger or exacerbate their MS and will need reassurance concerning this [6]. There is no clear evidence that vaccines increase the risk or the probability of MS or other CNS inflammatory demyelinating diseases [48,49]. People with MS should receive standard immunizations as recommended by the Ministry of Health, including the annual influenza vaccine. Live attenuated vaccines are contraindicated when immunosuppressive treatment has already started, but it is generally recommended to vaccinate if necessary before the start of immunosuppressive treatment [6,48,50]. In patients receiving immunomodulating treatments, namely interferons and GA, live attenuated vaccines can be given if clinically indicated [51].

An overview of the different types of vaccines is presented in Table 3. These include live-attenuated vaccines, which contain a weakened strain of the pathogen itself, which are potentially viable in immunocompromised patients. Patients with MS receiving immunosuppressive therapy should not receive a live-virus vaccine (for example., intranasal influenza (Flu-Mist), varicella zoster virus (Zostavax and Varivax), oral polio (Sabin vaccine), yellow fever, measles, rubella, mumps, typhoid, BCG). These patients should be immunized only in specific situations, on a case-by-case basis, and based on clinical needs and individual risk-benefit considerations. Other well-established types of vaccines that use non-viable pathogens, antigens isolated from pathogens, or toxins secreted by pathogens to generate a therapeutic immune response (for example., injected influenza, intramus-

cular polio (Salk vaccine), hepatitis A, hepatitis B, and tetanus) are considered safe for MS patients. More recently, mRNA vaccines have been developed for the management of COVID-19, associated with the SARS-CoV2 virus. These vaccines are the first examples of the use of an injected sequence of messenger RNA (mRNA) to generate antigens that give rise to immunity [52].

**Table 3.** Recommendations relating to vaccination of patients receiving specific disease-modifying therapies (DMT) for multiple sclerosis.

| Therapy | Impact According to Available Evidence | Recommendation |
|---|---|---|
| **Treatments that are unlikely to Impair the Effectiveness of Vaccinations** | | |
| Interferon beta [6] | Unlikely to impair the efficacy of vaccinations (data available for vaccination against influenza, pneumococcus, meningococcus, diphtheria-tetanus) COVID-19 vaccination immune response is likely to be intact [a] | Apply Saudi Ministry of Health recommendations without modification For the COVID-19 vaccine, no washout is required; vaccinate immediately |
| **Treatments with some concerns regarding their impact on vaccinations** | | |
| Glatiramer acetate | Response to influenza vaccine may be reduced vs. healthy controls or untreated MS patients (no data on other vaccines) | Apply Saudi Ministry of Health recommendations without modification |
| Dimethyl fumarate [53,54] | Risk of lymphopenia with dimethyl fumarate An open-label, multicenter study, demonstrated lower response rates to some vaccines in patients receiving dimethyl fumarate vs. interferon <br><br>• Td vaccination (68% vs. 73%) <br>• Diphtheria antitoxoid (58% vs. 61%) <br>• Pneumococcal serotype 3 (66% vs. 79%) <br>• Pneumococcal serotype 8 (95% vs. 88%) <br>• Meningococcal serogroup C (53% vs. 53%) <br><br>COVID-19 Vaccination immune response is probably intact [a] No data on other vaccines | Concomitant administration of non-live vaccines according to the MOH vaccination schedules may be considered during therapy. Do not administer live attenuated vaccines to patients undergoing therapy <br><br>For the COVID-19 vaccine, no washout is required; vaccinate immediately |
| Teriflunomide [53,55] | Teriflunomide-treated patients mounted appropriate immune responses to seasonal influenza vaccination (TERIVA study; >90% achieved post-vaccination antibody titers consistent with seroprotection)EMA, Bar-Or No data on other vaccines COVID-19 vaccination immune response is probably intact [a] | Apply vaccine recommendations for immunocompromised individuals [b] Do not administer live attenuated vaccines during and for at least 6 months after treatment. For the COVID-19 Vaccine, no washout is required; vaccinate immediately |
| Natalizumab [53,56] | Vaccine response to influenza is reduced kaufman EMA Little effect on levels of anti-Tetanus toxoid IgG antibodies or antibodies to keyhole limpet hemocyanin No data on other vaccines COVID-19 vaccination immune response is probably intact [a] | Apply vaccine recommendations for immunocompromised individuals [b] Do not administer live attenuated vaccines during treatment For the COVID-19 vaccine, no washout is required; vaccinate immediately |

**Table 3.** *Cont.*

| Therapy | Impact According to Available Evidence | Recommendation |
|---|---|---|
| Fingolimod [53,57,58] | Reduced response to vaccination vs. healthy controls, untreated patients, and patients on interferonβ: kappos ema Reduced responder rates for fingolimod vs placebo for influenza vaccine (54% vs. 85% at 3 weeks; 43% vs. 75% at 6 weeks post-vaccination) and for tetanus toxoid (40% vs. 61%; 38% vs. 49%, respectively)<br><br>COVID-19 vaccination immune response is mostly diminished [a] | Consider a complete vaccination schedule before starting the treatment<br>Avoid live attenuated vaccines during and for at least 2 months after discontinuation due to risk of infection. Other vaccines may not work as well as usual if given during this period<br>Assess patients for their immunity to varicella zoster virus (VZV)/(chickenpox) prior to treatment<br>Vaccinate for *varicella zoster* in antibody-negative patients at least one month before treatment.<br>For the COVID-19 vaccine, no washout is required to avoid MS rebound disease |
| Ocrelizumab [53,59,60] | Reduced vaccine response vs. healthy controls or patients on interferons for influenza, tetanus, pneumococcus However, patients on ocrelizumab mounted humoral responses to vaccination, although decreased vs. controls, for:<br>• Tetanus 24% vs. 55%<br>• Pneumococcal polysaccharide 23 valent 72% vs. 100%<br>• Seasonal influenza 56–80% vs. 75–97%<br>COVID-19 vaccination response could be weakened<br>No available data on other vaccines | It is not recommended to receive live-attenuated or live vaccines during treatment and after discontinuation until B-cell repletion<br>Immunization guidelines recommend a minimum of 6 weeks between immunization and treatment initiation,<br>For COVID-19 vaccines, vaccinate 3–4 months after the last dose of the DMT, with the second shot delayed by 2–4 weeks. Vaccination should be done ≥3 months after the last infusion. |
| Mitoxantrone | Vaccine response to influenza is reduced compared to healthy controls No available data on other vaccines | Apply vaccine recommendations for immunocompromised individuals [b] |
| **Treatments with insufficient or unavailable human data** | | |
| Alemtuzumab [53,61,62] | Not enough data to evaluate the vaccine's response<br>COVID-19 vaccination response is possibly diminished [a] | Patients should complete any necessary immunizations at least 6 weeks prior to treatment.<br>Do not administer live viral vaccines for at least 6 weeks before treatment, during treatment, or following a recent course of treatment<br>Prior to treatment, assess patients for immunity to *varicella zoster* virus (VZV). Consider vaccination for *varicella zoster* in antibody-negative patients and postpone alemtuzumab until 6 weeks post-vaccination.<br>For the COVID-19 vaccine, vaccination should be done only after the satisfactory recovery of lymphocyte counts Studies have shown that total lymphocytes remained below the lower limit of normal for around half a year after the first and second courses of treatment |
| Rituximab/other anti-CD20 [53,59,63] | No data on any vaccine for patients on rituximab or other anti-CD20 agents (other than ocrelizumab, see above) Response to inactivated vaccines may be reduced during and after treatment COVID-19 vaccination response is possibly diminished [a] | Vaccinations such as hepatitis vaccinations should be completed at least 4 weeks prior to the first administration of treatment.<br>Live virus vaccines should not be administered prior to or during treatment<br>For the COVID-19 vaccine, vaccinate 3–4 months after the last infusion. The second dose could be delayed by 2–4 weeks. Vaccination should occur ≥3 months after the last infusion. |

**Table 3.** *Cont.*

| Therapy | Impact According to Available Evidence | Recommendation |
|---|---|---|
| Cladribine [53,64,65] | No available data for any vaccine COVID-19 vaccination response is possibly weakened [a] | To allow for the full effect of vaccination to occur, administer all immunizations following guidelines with a minimum of 4–6 weeks after starting treatment. It is recommended to vaccinate patients who are antibody-negative for the varicella zoster virus before treatment initiation. Live-attenuated or live vaccines should be administered a minimum of 4–6 weeks prior to starting due to the risk of active vaccine infection As long as the patient's white blood cell counts are not within normal limits, vaccination with live or attenuated live vaccines should be avoided during and after cladribine treatment Timing of the COVID-19 vaccine in relation to cladribine treatment is not likely to significantly impact the vaccine response; therefore, the COVID-19 vaccine may be administered as soon as it is available to the patient any time after a course of cladribine (4 weeks gap is recommended). Resuming the next treatment course of cladribine should be 2–4 weeks after vaccine completion. |
| Siponimod [53,66] | No available data for any vaccine COVID-19 vaccination response is mostly weakened [a] | It is recommended to vaccinate patients who are antibody-negative for the varicella zoster virus before treatment initiation. Postponing treatment for at least 4 weeks to allow the full effect of vaccination to occur is recommended. Avoiding the use of live attenuated vaccines during treatment and for 4 weeks after discontinuing the treatment Other vaccines could be less effective if administered during treatment. It is recommended to discontinue treatment at least 1 week before and until 4 weeks after a scheduled vaccination. For the COVID-19 vaccine, vaccinate without washout, even if the response is possibly diminished, to avoid MS rebound disease |
| Methotrexate | No data for any vaccine | Apply vaccine recommendations for immunocompromised individuals [b] Avoid vaccination with live vaccines |
| Cyclophosphamide | No data for any vaccine | Apply vaccine recommendations for immunocompromised individuals [b] |

[a] Expert opinions based on the mechanisms of action of DMTs [b] According to recommendations from the French Multiple Sclerosis Society 30: live attenuated vaccines are contraindicated; give recommended vaccines as per the vaccination schedule for the general population and vaccines specifically recommended in immunocompromised patients (influenza and Pneumococcus in particular); based on "expert recommendation." Collated from references [6,49,51] except where stated in the table. * COVID19: Coronavirus disease.

### 3.2. Potentially Immunosuppressive Therapies and Vaccination

The approach to vaccination of a person with MS may vary to some extent according to the effects on the immune system of the individual DMT. Table 4 summarizes the current knowledge on the impact of DMTs on the efficacy of vaccinations and presents practical recommendations on vaccination for these patients [6,48,51,52,54–56,62,63,65,67–72].

**Table 4.** Examples of vaccine types compiled from information presented in references [52,67].

| Type | Examples |
|---|---|
| Live-attenuated vaccines | Measles, mumps, rubella (MMR combined vaccine)<br>Rotavirus<br>Smallpox<br>Chickenpox<br>Yellow fever |
| Inactivated vaccines | Hepatitis A<br>Influenza (injection)<br>Polio (injection)<br>Rabies |
| Subunit, recombinant, polysaccharide, and conjugate vaccines | Hib (Haemophilus influenzae type b) disease<br>Hepatitis B<br>HPV (Human papillomavirus)<br>Rubella (part of the DTaP combined vaccine)<br>Pneumococcal disease<br>Meningococcal disease<br>Varicella zoster (shingles) |
| Toxoid vaccines | Diphtheria<br>Tetanus |
| mRNA vaccines | SARS-CoV2 |

Given that patients with MS receiving immunosuppressive/immunomodulating DMTs may be at increased risk of infection, it is likely that the potential benefits of vaccination outweigh any potential risks. Interferon beta and glatiramer acetate, dimethyl fumarate, teriflunomide, and natalizumab are unlikely to diminish the efficacy of subsequent vaccination to an important extent compared to sphingosine-1-phosphate (S1P) receptor modulators, cladribine, alemtuzumab and anti-CD20 therapies that could diminish vaccine efficacy [9,51,73,74]. Initiation of treatment with these DMTs should be delayed for 4–6 weeks after completion of the required vaccinations (in most cases, see Table 2 for details). Data are lacking for alemtuzumab, cladribine tablets, and anti-CD20 agents other than ocrelizumab (rituximab), siponimod, methotrexate, and cyclophosphamide. Administration of these DMTs should be delayed until after the completion of vaccinations, as described above. Patients treated with glatiramer acetate, dimethyl fumarate teriflunomide, natalizumab, mitoxantrone, methotrexate, and cyclophosphamide should be managed as immunocompromised patients, according to recommendations from the French Multiple Sclerosis Society [51].

It is also important to screen for potential opportunistic infections, as described in the labeling of individual DMTs, before starting treatment and to vaccinate as soon as possible. This will help to reduce delay in prescribing immunosuppressive/immunomodulating DMT if this is required later.

### 3.3. Special Considerations with Respect to the Seasonal Influenza Vaccine in the MS Population

The FluMist (nasal spray) vaccine is a live vaccine; therefore, it is not recommended [48]. Moreover, there is inadequate evidence to support or negate any correlation between influenza vaccination and MS exacerbations. In contrast, there may be an association between influenza infection and MS exacerbations and an increased impact of influenza infections on chronic diseases in general. Therefore, influenza vaccination provides benefits that exceed possible risks. Clinicians must also take into consideration that it is likely that fingolimod, mitoxantrone, alemtuzumab, cladribine, ocrelizumab, and rituximab can reduce the body's ability to mount a sufficient response after an influenza vaccine.

### 3.4. Special Considerations with Respect to the COVID-19 Vaccination in the MS Population [53,58,75]

Several vaccines against SARS-CoV2 are already approved and available for use, such as Pfizer/BioNTech, Moderna, Oxford University/AstraZeneca, and Sputnik V.

Based on currently available evidence, the benefits of all approved COVID-19 vaccines prevail over any possible risks of vaccination. In addition, there are no reports on the specific adverse effects of COVID-19 vaccines on people with MS. Therefore, MS patients should be reassured and encouraged to receive any of the approved vaccines whenever available.

In regard to its administration, COVID-19 vaccination should be completed at least 2–4 weeks before the initiation of a new disease-modifying therapy to ensure the best vaccine response, and patients should be encouraged to receive a booster dose of vaccine after consulting with their neurologists, particularly if they are receiving anti-CD20 therapy.

If patients are on Fingolimod therapy, it is advisable to receive the booster vaccination before initiating therapy. If this is not possible, the healthcare practitioner should assess the need for a booster dose on a case-by-case basis with the possibility of switching to another DMT to mount adequate immunity to SARS-CoV-2. As for ongoing treatment with interferons, glatiramer acetate, dimethyl fumarate, teriflunomide, or natalizumab, COVID-19 vaccination should be administered immediately without any adjustments or alterations to the DMT administration or schedule. Furthermore, for ongoing treatment with alemtuzumab and cladribine (which are known to induce profound reductions in circulating immune cells), COVID-19 vaccination should be administered after adequate recovery of lymphocytes after the treatment cycle (24 weeks or more after the last alemtuzumab dose) [59,61,64]. However, according to the last updated guidance from the United Kingdom National Health Service and the United States National MS Society (US NMSS), the limited available data do not suggest that the timing of the vaccine in relation to cladribine treatment is likely to impact the vaccine response. Therefore, the COVID-19 vaccine may be administered as soon as it is available to the patient at any time after a course of cladribine (a 4-week gap is recommended). Resuming the next treatment course of cladribine should be 2–4 weeks after vaccine completion. In regard to treatment with anti-CD20 agents, such as ocrelizumab, COVID-19 vaccination may be administered 3–4 months after the last dose of the anti-CD20 agent due to their potential to cause immune reconstitution syndrome [53]. For ongoing treatment with fingolimod or other DMTs sharing its mechanism and due to the risk of rebound of disease activity if withdrawn, COVID-19 vaccination may be administered without any adjustments or alterations to the DMT administration or schedule. However, note that a diminished immune response to vaccination has been observed in patients receiving fingolimod [58].

If a DMT was withdrawn to allow for COVID-19 vaccination, the recommendation is to wait 2–4 weeks before restarting/resuming the DMT to ensure the best vaccine response. However, it is not recommended to withdraw DMT for COVID-19 vaccination due to the risk of relapse. Given the current observed suboptimal immune response to the standard one- or two-dose schedule in people receiving DMTs, an additional dose of the COVID-19 vaccine may be beneficial [51].

Overall, more studies are needed to monitor humoral and cellular immune responses to COVID-19 vaccines in people with MS using some DMTs. Additional research is needed to establish the optimal COVID-19 vaccine schedule for this population.

### 3.5. Vaccination and MS Relapses

Vaccination should not be delayed because of relapse or due to steroids. However, vaccination should be delayed for ≥3 months for patients who have received long-term high-dose steroids (≥20 mg prednisone) for more than 2 weeks [76].

## 4. Conclusions

The appropriate diagnosis and management of different MS symptoms require an in-depth understanding of these symptoms, as highlighted in this review. MS patients should be encouraged to receive a vaccination if the contraindications described in this review are not met.

**Author Contributions:** Conceptualization, I.A.A.T. and M.A.A.J.; Methodology, I.A.A.T. and M.A.A.J.; Writing—Original draft preparation, I.A.A.T., M.M.A. and M.A.A.J.; Writing—Review and Editing, I.A.A.T., M.M.A. and M.A.A.J.; Validation and Review, N.A.F., D.A.A., A.H.A.-J., Y.M.A.M., M.A.A., H.Y.A.-M., B.A.A.-O., H.S.A., R.H.A.Y., M.A.B., R.F.B., E.J.C., M.H., H.M.K., S.M., L.H.S., J.A.S., E.S. and M.A.A.J. All authors have read and agreed to the published version of the manuscript.

**Funding:** This project was funded by the Ministry of Health, Kingdom of Saudi Arabia.

**Institutional Review Board Statement:** Not applicable.

**Informed Consent Statement:** Not applicable.

**Data Availability Statement:** Not applicable.

**Conflicts of Interest:** The following authors declared no conflicts of interest regarding the publication of these recommendations: Dema A. Alissa, Bedor A. Al-Omari, Hessa Sharar AlOtaibi, Mohammed Hakami, Leena H. Saeed, Ahmed H. Al-Jedai, Hajer Al-Mudaiheem.

**Disclosure:** Ibtisam A. Al Thubaiti received a speaker honorarium from Novartis and Merck Serono and received consultancy fees from Merck Serono. Mona M. AlKhawajah received a speaker honorarium, consultancy fees, or travel support from Roche, Merck, Sanofi, Biogen, Novartis, Saja, Hikma, and Actelion. Norah Al Fugham received travel support from Sanofi Genzyme. Yaser M. Al Malik received speaker honoraria from Merck and Roche, received consultancy fees from Merck, Genzyme, Novartis, and Roche, and received travel support from Roche, Biogen, and Serono. Mousa A. Almejally received speaker honoraria from Sanofi and Biogen and received travel support from Saja, Novartis, Biogen, and Merck. Rumaiza Hussein Al Yafeai received speaker honoraria from Novartis and Roche. Mohammed Ahmed Babakkor received travel support from Bayer. Reem F. Bunyan received speaker honoraria and travel support from Merck, Novartis, and Roche. Edward J. Cupler received speaker honoraria from Novartis, Biogen, Sanofi, and Merck and received travel support from Novartis, Biogen, Sanofi, and Merck. Hanaa Mohamed Kedah received speaker honoraria from Novartis, Biogen, and Merck and received travel support from Merck and Bayer. Seraj Makkawi received a speaker honorarium from Merck and a speaker honorarium and travel support from Roche. Jameelah A. Saeedi received speaker honoraria and/or consultancy fees or travel support from Roche, Novartis, Merck, Hikma, Biologix, Sanofi, and Bayer. Eslam Shosha received speaker honoraria from Biologix, Hikma, and Merck, received consultancy fees from Merck and Sanofi, and received travel support from Biologix, Merck, Sanofi, and Roche. Mohammed A. Al Jumah received consultancy fees or speaker honoraria from Merck, Biogen, Biologix, Novartis, Sanofi, Bayer, and Roche and received research grants from Merck.

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
