# Peer review of "Saudi Consensus Recommendations on the Management of Multiple Sclerosis: Symptom Management and Vaccination"

_ctn, doi:10.3390/ctn7010006_

Round 1

Reviewer 1 Report

The article titled “Saudi consensus recommendations on the management of Multiple Sclerosis: Symptom management and vaccination” discusses the latest recommendations for managing symptoms of MS and vaccinating patients receiving disease-modifying therapies (DMTs). The paper is well written, and the writing is clear and concise. It presents an updated overview of different therapies that are being used in the treatment and management of the most common MS symptoms. These recommendations serve as an excellent guide for clinicians treating MS patients. For this reason, I recommend the publication of this work with the following minor comments that I believe would make this already well-written work even more informative.

MS is 2 to 3 times more common in women compared to men. However, the authors have not discussed how certain drugs used to treat MS symptoms could affect pregnancy and the fetus. The rising prevalence of MS in women is still a clinical concern, as several of the available DMTs may have undesirable consequences and therefore not compatible with pregnancy. This might be a topic on its own, but I believe if authors add a brief comment on which of the drugs used to treat MS symptoms are compatible with pregnancy, it will contribute to the improvement of this study.

Overall, I recommend the publication of these updated recommendations which are highly beneficial for clinical practices. 

Author Response

Thank you for your comments. Please see attachment for response. 

Reviewer 2 Report

In this manuscript, Ibtisam A. AlThubaiti et al propose a consensus recommendations for MS symptom management and for vaccination in Multiple Sclerosis . The manuscript is interesting. However, the two parts are not well articulated, not written in the same manner and it is not clear why the authors propose to address those two topics in the same review. Maybe two separate manuscripts would be more appropriate. In addition, they discuss the symptoms management somewhat superficially and it would be advised to add illustrations and tables to support the recommendations.

Major comment:

In the first part, the authors discuss the management of several symptoms and discuss more in detail the management of Fatigue before listing the one of other symptoms. Except for Fatigue, the other topics are discussed succinctly.

In particular, for the urinary tract symptoms/bladder dysfuntion, the authors should discuss the different approaches regarding the symptoms in more details. Bowel dysfunction should also be included as it is related to bladder dysfunction. Indeed bladder and bowel dysfunction should be discussed together with sexual dysfunction. Thus “the symptoms” should be ordered in a more coherent manner.

For the urinary symptoms, the authors propose that anticholinergic drugs may be considered as first-line option but it is not specified for which bladder dysfunction this should be proposed. In addition, adverse effects of anticholinergic drugs (eg. Confusion) should be discussed.

It would be important to elaborate on treatment strategies also for patients with failure to empty the balder (for example with alpha antagonists) or patients with detrusor overactivity. An illustration of the bladder innervation would be useful for a better understanding and a table with a review of the different drugs used to treat balder dysfunction would be useful.

Concerning the second part on vaccinations:

The authors propose that live attenuated vaccine are contraindicated under an immunosuppressive treatments. What is their recommendations under first line platform DMT, in particular Interferons and Glatiramere acetate? This is not clear in the text.

The discussion on vaccination resembles a list of bullet points. It would be better to have it presented in the same manner as Part 1 on symptoms.

As the authors discuss COVID vaccination in detail, it would be interesting to also discuss early anti-viral treatment and preventive treatment, especially under anti-CD20 treatments. The natural course of a COVID infection in MS should also be discussed in light of the recent epidemiological studies.

Author Response

Thank you for your comments. Please see the attachment for responses. 

Round 2

Reviewer 2 Report

The manuscript has been improved and is more coherent in the present form.

Minor comments:

It should be clearly highlighted that this manuscript is complementary to the previous published manuscript from the same authors on DMTs (Clin. Transl. Neurosci. 20226(4), 27; https://doi.org/10.3390/ctn6040027).

The paragraph on Pregnancy is neither related to the symptom treatment nor to the vaccination but to the use of DMT during Pregnancy. It is too succinct to bring sufficient information on the subject. It should be removed.

Author Response

Dear Reviewer,

As per your comment, it was highlighted that this manuscript is complementary to the previously published manuscript from the same authors on DMTs. 

The paragraph on Pregnancy was also removed.

Thank you for your comments.